# Genome-Wide Association Study Identifies a Rice Panicle Blast Resistance Gene *Pb3* Encoding NLR Protein

**DOI:** 10.3390/ijms232214032

**Published:** 2022-11-14

**Authors:** Lu Ma, Yao Yu, Changqing Li, Panting Wang, Kunquan Liu, Wenjing Ma, Wei Wang, Yunxin Fan, Ziwei Xiong, Tingting Jiang, Jingran Zhang, Zhixue Wang, Jianfei Wang, Hongsheng Zhang, Yongmei Bao

**Affiliations:** 1State Key Laboratory of Crop Genetics and Germplasm Enhancement, College of Agriculture, Jiangsu Collaborative Innovation Center for Modern Crop Production, Cyrus Tang Innovation Center for Crop Seed Industry, Jiangsu Province Engineering Research Center of Seed Industry Science and Technology, Nanjing Agricultural University, Nanjing 210095, China; 2School of Life Sciences and Biotechnology, Shanghai Jiao Tong University, Shanghai 200240, China; 3School of Agriculture and Biology, Shanghai Jiao Tong University, Shanghai 200240, China

**Keywords:** rice blast, panicle blast, GWAS, haplotype analysis, resistance gene, *Pb3*

## Abstract

Rice blast is a worldwide fungal disease that seriously affects the yield and quality of rice. Identification of resistance genes against rice blast disease is one of the effective ways to control this disease. However, panicle blast resistance genes, which are useful in the fields, have rarely been studied due to the difficulty in phenotypic identification and the environmental influences. Here, *panicle blast resistance-3* (*Pb3*) was identified by a genome-wide association study (GWAS) based on the panicle blast resistance phenotypes of 230 Rice Diversity Panel I (RDP-I) accessions with 700,000 single-nucleotide polymorphism (SNP) markers. A total of 16 panicle blast resistance loci (PBRLs) within three years including one repeated locus PBRL3 located in chromosome 11 were identified. In addition, 7 genes in PBRL3 were identified as candidate genes by haplotype analysis, which showed significant differences between resistant and susceptible varieties. Among them, one nucleotide-binding domain and Leucine-rich Repeat (NLR) gene *Pb3* was highly conserved in multiple resistant rice cultivars, and its expression was significantly induced after rice blast inoculation. Evolutionary analysis showed that *Pb3* was a typical disease resistance gene containing coiled-coil, NB-ARC, and LRR domains. T-DNA insertion mutants and CRISPR lines of *Pb3* showed significantly reduced panicle blast resistance. These results indicate that *Pb3* is a panicle blast resistance gene and GWAS is a rapid method for identifying panicle blast resistance in rice.

## 1. Introduction

Rice blast, caused by the fungi *Magnaporthe oryzae*, is one of the most serious and recurrent diseases in rice-growing regions worldwide [1]. It is estimated that every year, the disease-destroyed rice could feed 60 million people [2,3]. Breeding new resistant varieties with resistance (R) genes are the most effective way to solve the problem [4]. To date, more than 100 rice blast resistance (R) genes have been identified and 35 of them have been cloned [5]. However, panicle blast resistance genes have rarely been studied due to the difficulty in phenotypic identification and the environmental influences [6]. *Panicle blast resistance-1* (*Pb1*) was the first cloned panicle blast resistance gene encoding a coiled-coil-nucleotide binding site-leucine rich repeat (CC-NBS-LRR) protein, which was isolated from the indica cultivar Modan and conferred durable and broad-spectrum resistance to rice blast [7]. *Pi25* was derived from the indica rice variety Gumei 2 and provided both panicle blast and leaf blast resistance [8]. The *Pigm* contains two NBS-LRR proteins, *PigmR* and *PigmS*, which endow rice with blast resistance and yield balance [9]. *Pb2* is an NLR gene located at the end of chromosome 11 identified by GWAS and complementation transgenic plants phenotype analysis confirmed that it has resistance to panicle blast and leaf blast resistance [10].

Among the 35 cloned resistance genes, 31 genes encode the nucleotide binding-leucine rich repeats (NBS–LRR) protein [11]. The NBS domain contains three highly conserved kinase structures that bind to phosphoric acid to obtain energy for defense against pathogens, and the C-terminal LRR participates in protein–protein interactions of R and Avr-genes [11]. *Pib* is the first NBS-LRR-type rice blast resistance gene that was cloned in 1999 [12]. The NLR gene is often repeated in tandem to regulate rice blast resistance [13]. For example, *Pi2*, *Pi9*, *Pi50*, *Pigm*, and *Piz-t* are all located in the pericentric region of chromosome 6, belonging to the same genetic locus [5]. In addition, *Pi-d2*, *pi21, Ptr*, and *bsr-k1* encode receptor-like kinase, proline-rich protein, atypical four Armadillo repeats protein, and tetratricopeptide repeats protein, respectively [14,15,16,17]. In the field cultivation, Blast resistance mediated by single or a few major blast resistance genes (R genes) is easily overcome by *M. oryzae*. However, partial resistance maintains continuous field resistance, which is controlled by minor polygenes or the quantitative trait locus (QTL) [18]. Therefore, identifying new resistant QTLs from rice germplasms is an effective way to control rice blast [19].

Genome-wide association study (GWAS) is a genome-wide genetic marker detection technology, which has been developed to be an important method to mine the inheritance of complex diseases [20]. Compared with traditional bi-parental linkage analysis, GWAS has a high efficiency to identify QTLs in natural varieties [21,22]. In 2005, GWAS was first applied to human disease age-related macular degeneration [23]. With the advent of next-generation sequencing technology, a large amount of genotype data has been obtained in plants, including single-nucleotide polymorphisms (SNPs), insertions and deletions (INDEL), and structure variation (SV), and these played an important role in the development of the GWAS method. In the past 10 years, GWAS has grown into a popular and powerful tool to mine genes/QTL for complex agricultural traits [24]. In wheat, 160 resistance loci associated with stem rust resistance including 21 newly discovered loci were identified by GWAS with 283 durum wheat varieties and 26,439 SNP markers [25]. In maize, 39 SNP loci associated with fumonisin resistance were identified by GWAS by utilizing GBS (Genotyping-By-Sequencing) technology to genotype 270 inbred lines [26]. In soybean, 30 loci associated with flowering and maturation time, and plant height and yield were detected by GWAS based on the phenotype data of 113 germplasms and 4442 high-quality SNP markers [27].

GWAS has also been widely used to study rice complex traits such as yield and disease resistance. With 28,445 SNP markers by resequencing 115 rice japonica cultivars, 16 loci associated with amylose content and resistant starch were identified by GWAS [28]. GWAS analysis of 1,345,417 SNPs on the whole genome of 950 widely collected rice varieties detected 10 QTLs associated with yield traits [29]. With 2,888,332 high-confidence SNPs of 259 rice varieties, 63 BLB (bacterial leaf blight) resistance loci containing 954 genes were identified by GWAS [30]. Ten genomic regions (GRs) with 147 single-nucleotide polymorphisms associated with Rice black-streaked dwarf virus (RBSDV) resistance were identified by GWAS with 1956 accessions from the 3000 Rice Genomes Project (3K RGP) [31]. Rice diversity panel 1 (RDP-1), consisting of 413 *O. sativa* cultivars collected from 82 countries, is publicly available and widely applied to identify resistance-related loci by GWAS [32,33]. By using 162 rice cultivars of RDP-1, 31 QTLs associated with blast resistance were identified by inoculating eight rice blast isolates from four African counties [34]. Totals of 373, 356, and 336 rice cultivars in RDP-1 were evaluated in the field in three major rice production areas of China, and 74 candidate genes in the region of 16 LAFBRs were identified by GWAS [33]. Combining 362 rice cultivars of RDP-1 and 700K high-density genetic markers, 97 loci were identified associated with blast resistance (LABRs) against five *M. oryzae* isolates [35]. The Rice Diversity Panel II (RDP-II), which contains 584 rice accessions and is genotyped with 700K SNPs, was inoculated with three *M. oryzae* isolates. Then, 27 loci associated with rice blast resistance (LABRs) were detected by GWAS, and a partial resistance gene *PiPR1* was identified [36]. *Pb2* encodes an NBS-LRR protein, identified in the GWAS of 230 cultivars of RDP-1, which enhanced the resistance of panicle blast and leaf blast without affecting agronomic traits and can be used for rice blast resistance breeding [10]. GWAS analysis was carried out by using broad-spectrum resistant rice variety Digu and 66 no broad-spectrum resistant accessions, and found that the natural variation of *bsr-d1* promoter altered resistance to rice blast [37]. *bsr-d1* is a C2H2-type transcription factor that is regulated by the MYB transcription factor and enhances blast resistance by inhibiting H_2_O_2_ degradation, and has no obvious effect on rice yield and quality. Therefore, *bsr-d1* has a broad application prospect in rice breeding for blast resistance.

In this study, we evaluated the panicle blast resistance of 230 cultivars derived from RDP-1 by field injection inoculation with the *M. oryzae* strain 2014-290-B27. By GWAS analysis of population phenotypes and 700K SNPs, 16 panicle blast resistance loci (PBRL) within three years including one repeated locus PBRL3 located in chromosome 11 were identified. In addition, 7 genes in PBRL3 were identified as candidate genes by haplotype analysis, which showed a significant difference between resistant and susceptible varieties. Among them, one Nucleotide-binding domain and Leucine-rich Repeat (NLR) gene *Pb3* was highly conserved in multiple resistant rice cultivars, and its expression was significantly induced after rice blast inoculation. Compared with the wild-type, the T-DNA insertion mutant lines and CRISPR knockout transgenic lines showed significantly reduced resistance to leaf blast and panicle blast. In summary, this study identified a Nucleotide-binding domain and Leucine-rich Repeat (NLR) panicle blast resistance gene *Pb3* by GWAS analysis, which can significantly improve rice panicle blast and leaf blast resistance.

## 2. Results

### 2.1. Phenotype Characteristics of Panicle Blast Resistance in the Natural Population

The natural population used in this study consisted of 230 rice accessions of RDP-1, including 43 indica (IND), 51 tropical japonica (TRJ), 51 temperate japonica (TEJ), 7 aromatic (ARO), 45 aus (AUS), and 34 area mixture (ADM) (Figure 1A). A neighbor-joining analysis was constructed to understand the genetic diversity of 230 accessions used in this study and classified the population into six groups (Figure 1B). To evaluate the panicle blast resistance of the 230 accessions, we inoculated the population with blast strain 2014-290-B27 and evaluated the percentage of diseased grains for each accession in the field in 2017, 2018, and 2019 (Figure 2A–C). Based on the phenotype in three experiments, we analyzed the differences in panicle blast resistance among sub-populations of RDP-1. In 2017, the percentages of diseased grains were 35.57%, 17.10%, 51.08%, 26.00%, 56.15%, and 26.43% in IND, AUS, ADM, ARO, TEJ, and TRJ, respectively (Figure 2D). In 2018, the percentages of diseased grains of IND, AUS, ADM, ARO, TEJ, and TRJ were 11.67%, 6.68%, 35.12%, 4.14%, and 40.87%, respectively (Figure 2E). In 2019, the percentages of diseased grains of IND, AUS, ADM, ARO, TEJ, and TRJ were 28.44%, 16.64%, 44.71%, 14.14%, and 55.59%, respectively (Figure 2F).

### 2.2. Identification of PBRLs by Using GWAS

We performed GWAS for the three experiments with the Efficient Mixed-Model Association eXpedited (EMMAX), which accounts for both population structure and kinship [38]. The quantile–quantile plots showed that the model of the three environments could be used to identify association signals (Appendix A). A total of 16 panicle blast resistance loci (PBRLs) mainly distributed on chromosomes 6 and 10 were identified by GWAS with high-quality genetic markers filtered from 700K SNPs (Figure 3A–C). Among the 16 PBRLs, 9 loci were co-localized with previously mapped or cloned R genes or loci (Table 1). PBRL1 was located at the terminal of chromosome 2 and co-localized with the previously identified *Pig(t)* [39]. PBRL6 was located in the 20 Mb region of chromosome 4 and contained two reported resistance genes *Pi-21* or PBRL-14 locus [10,15]. Among the 16 PBRLs, PBRL3 was the only repeated locus detected both in 2017 and 2019, and it was located in the *Pik* locus at the end of chromosome 11 [40].

### 2.3. Analysis of Candidate Genes for Panicle Blast Resistance

To identify candidate resistance genes underlying the stable PBRL3 locus, we investigated the haplotype block structure based on locus within 500 kb intervals; however, it was difficult to find the LD block in the interval (Figure 4A–C). According to genome-wide LD, decay rates of indica and japonica were estimated at ∼200 kb [41], and a 200 kb interval (Chr11:27.1 Mb–27.3 Mb) was selected to predict candidate genes. With reference to the Nipponbare genome sequence and gene annotation, 15 candidate genes with SNPs on the promoter or exons region were obtained. Haplotype analysis results showed that 7 candidate genes with different haplotypes had significant phenotypic differences in the natural population and were selected as potential candidate genes for further analysis (Figure 5A). Among them, *LOC_Os11g44990* and *LOC_Os11g45090* encode CC-NBS-LRR proteins, *LOC_Os11g44910* encodes a DEAD-box RNA helicase, *LOC_Os11g44930* encodes pentatricopeptide repeat domain protein, *LOC_Os11g44950* encodes a glycosyl hydrolase, and *LOC_Os11g44890* and *LOC_Os11g45030* encode expressed proteins (Table 2). To investigate expression patterns of candidate genes after *M.oryzae* infection, we performed quantitative real-time PCR (qRT-PCR) analysis of resistant japonica landrace Bodao after seedling blast inoculation. The results showed that *LOC_Os11g44910*, *LOC_Os11g449300*, *LOC_Os11g44950*, *LOC_Os11g44990,* and *LOC_Os11g45090* were significantly induced after inoculation. Among them, *LOC_Os11g45090*, which was most significantly induced by rice blast fungi and encodes NBS-LRR domain protein, was reported to be involved in rice blast resistance [42]. We concluded that *LOC_Os11g45090* was the candidate gene in PBRL3 and then was named *Pb3* (Figure 5B).

### 2.4. Pb3 Is an R Gene That Regulates Panicle Blast Resistance

*Pb3* encodes a protein with a CC-NBS-LRR domain. In the high-quality 700K SNPs dataset, six SNPs causing amino acid changes were found in exons. In addition, SNP11-27282578 was located on the coiled-coil domain, SNP11-27282896 was located on the predicted NB-ARC domain, and SNP11-27284006 and SNP11-27285004 were located on the LRR domains. Moreover, the variation in the SNP11-27285004 in natural populations (CAG to TAG) leads to premature termination of *Pb3* protein translation, suggesting that the terminal LRR domain may be involved in resistance to rice blast. Haplotype analysis showed that there were 5 haplotypes of *Pb3* in the natural population (Figure 6A,B). The Hap.A haplotype was significantly more resistant to panicle blast than Hap.C and Hap.D (Figure 6C,D). The resistance haplotype A mainly contains TEJ cultivars, while the susceptible haplotypes C and D include 20% indica and 60% japonica (Figure 6E). To study the genetic diversity of *Pb3* in natural populations, we calculated the nucleotide diversity (π value) of the chromosome 11 region of 230 varieties by VCF tools, and the results showed that the π value of the *Pb3* locus was 0.0002, which was significantly lower than the average nucleotide diversity of the chromosome 11 (π value = 0.0003), illustrating the sequence conservation of *Pb3* in natural populations (Figure 6F). We carried out evolutionary analysis on the cloned rice blast R gene, and the results showed that *Pb3* was a typical NLR gene, which was highly homologous to *Pit* and was predicted to play similar functions (Figure 7A,B).

To investigate whether *Pb3* was involved in rice blast resistance, we obtained T-DNA insertion lines (pb3) from Salk Institute Genomic Analysis Laboratory (Figure 8A). PCR analysis confirmed T-DNA insertions in the exon region (Appendix A). Then, we inoculated mutant and wild-type Dongjin at the booting and seedling stages, respectively. Blast vaccination results showed that the average lesion length and number of *Pb3* were significantly higher than those of the wild type (Figure 8B,C,E). Moreover, 5 mutants were more susceptible to the *M. oryzae* strain than WT by blast inoculation at the booting stage (Figure 8D,F). To further verify the rice blast resistance of *Pb3*, we knocked out this gene in the Bodao background with the CRISPR/Cas9 gene editing system. A total of 3 homozygous CRISPR lines (type1, type2, and type3) were obtained (Figure 9A,B). The mutation target was located on the LRR domain at the end of the gene. Inoculation of the booting stage results showed that 5 mutants of 2 lines were more susceptible than WT to the *M. oryzae* strain (Figure 9C,D). It showed that the LRR domain confers resistance to rice blast in *Pb3*, which was consistent with the results of haplotype analysis. These results further validated that *Pb3* was a new R gene in the *Pik* locus that regulates both leaf blast and panicle blast resistance in rice.

## 3. Discussion

Rice blast is one of the critical rice diseases, and leaf blast and panicle blast are the most common types of this disease. Compared with leaf blast, panicle blast is considered to be more destructive, which can cause direct yield losses up to 70% or even 100% in fields by affecting grain sterility, rotting the branch and neck, and even losing the entire panicle [43,44]. Isolating the panicle blast resistance gene and introducing it into the susceptible elite cultivars is the most economical and effective way to control rice blast disease. Blast resistance is generally classified into complete and partial resistance [45]. Complete resistance to blast is qualitative and race-specific and is usually controlled by a major gene [4]. However, this type of resistance is easily overcome by the variety of races in the field. Partial resistance controlled by minor polygenes or the quantitative trait locus (QTL) confers broad-spectrum and durable resistance to *M. oryzae*.

The Rice Diversity Panel (RDP-1) contains substantial genetic and phenotypic diversity and has been used for GWAS analysis of various traits in rice [46]. We selected 230 cultivars in RDP-1 for panicle blast resistance identification in the field. Three experimental field panicle blast inoculations were performed on these cultivars. The average diseased grains percentages were 23.33%, 12.53%, 50.37%, and 38.70% in IND, AUS, TEJ, and TRJ, respectively. These results are consistent with previous studies where the TRJ and IND subgroups of rice were more resistant than the TEJ subgroups [33]. We chose the EMMAX model for association analysis and identified a repetitively mapped locus PBRL3 within the *Pik* locus at the end of chromosome 11. Haplotype analysis and expression pattern analysis identified 5 candidate genes that were significantly induced by *M. oryzae*. Among them, *LOC_Os11g44910* encodes a DEAD-box RNA helicase, which contains a conserved amino acid sequence Asp-Glu-Ala-Asp (DEAD). DEAD box RNA helicase family proteins can hydrolyze ATP to obtain double-stranded energy unwinding RNA, and are reported to be involved in plant leaf growth and the cold stress response [47,48]. *LOC_Os11g44930* encodes a pentatricopeptide-repeat (PPR)-domain-containing protein. Plant PPR proteins are usually located in mitochondria or chloroplasts, which edit and modify the RNA transcripts to regulate plant growth and development [49,50]. *LOC_Os11g44950* is a member of the glycosyl hydrolase family 3. Glycosyl hydrolase can effectively hydrolyze sugar-based substances and are involved in cell wall remodeling [51]. *LOC_Os11g44990* and *LOC_Os11g45090* are NBS-LRR-type genes that are widely reported to be involved in rice blast resistance [52]. Furthermore, after inoculation with *M. oryzae*, the expression of *LOC_Os11g45090* significantly increased at 24 h by 4.07-fold. These results indicate that *LOC_Os11g45090* (*Pb3*) is the most likely candidate gene in PBRL3.

In the 230 natural populations, five SNPs causing amino acid changes were detected in the coding region of *Pb3*. Among them, SNP11-27282578 fell on the CC domain, SNP11-27282896 was located in the NB-ARC domain, and SNP27284006 and SNP11-27285004 fell on the LRR domain. Combining the haplotype analysis results, we found that the difference between resistant haplotype Hap. A and susceptible haplotypes Hap. C and Hap. D contained these four SNPs that were located in the functional domain of *Pb3*. In NBS-LRR proteins, the CC domain mainly mediates the interaction with transcription factors and structural proteins [53]. NBS has the function of ATP or GTP binding and hydrolysis, which can resist the invasion of pathogens by gaining energy [11]. The easily altered structure of LRR may be related to its ability to specifically recognize pathogen-effector proteins [54]. To verify the function of these domains, we obtained the transgenic plants of *Pb3*. We first obtained a T-DNA insertion mutant of *Pb3*, and the insertion position was located in the double coiled-coil region. The susceptible phenotype of the mutant line verified that *Pb3* was indeed involved in rice blast resistance. The genotype data revealed an SNP11-27285004 on the LRR domain in the end of *Pb3* in natural populations, which can prematurely terminate the translation of the protein, resulting in the deletion of the LRR domain. Therefore, we specifically constructed a CRISPR-Cas9 mutation target located on the terminal LRR domain, and CRISPR showed that it was more susceptible than wild-type thin rice, indicating the importance of the LRR domain for *Pb3* to play the role of panicle blast resistance.

Most of the cloned rice blast resistance genes encode NBS-LRR domain proteins. Therefore, we performed phylogenetic tree analysis on *Pb3* and cloned R genes and found that Pit has the highest homology with *PB3*. Pit interacts with OsRac1 through its NB-ARC domain and activates OsRac1, which plays an important role in Pit-mediated immune responses [55]. Therefore, *Pb3* may also mediate resistance by binding to a key immune regulatory factor. In the following studies, we can verify the interaction between *Pb3* and OsRac1, and can also search for other interacting proteins in combination with yeast two-hybrid assays to research the specific resistance mechanism of *Pb3*. On the other hand, we verified the function of the LRR domain at the end of *Pb3* through the CRISPR plants. The deletion of the LRR domain may affect the recognition of effector proteins of rice blast fungi, further inhibit the activation of the rice defense response, and reduce rice resistance to rice blast. In summary, we identified a repeat site PBRL3 on chromosome 11 by GWAS. Through functional verification of candidate genes, we found that *Pb3*, which encodes NBS-LRR protein, is a causal gene in PBRL3, which can regulate the resistance of rice to leaf blast and panicle blast. In this work, we carried out GWAS analysis to quickly and accurately obtain the rice blast resistance R gene, which provides a theoretical basis for mining a large number of panicle blast resistance genes. With the continuous development of sequencing technology, it will be more accurate to locate panicle blast resistance genes through GWAS in the future. However, due to the influence of environmental factors, identification of resistance to panicle blast in the field is still a difficult problem, which is crucial to the results of GWAS analysis.

## 4. Materials and Methods

### 4.1. Plant Materials and Fungal Growth

RDP-1, which contains 413 *O. sativa* accessions from 82 countries, was provided by Cornell University [46]. According to the growth period of these accessions, we selected 230 accessions from the panel to ensure the growth period consistency of rice. The whole panel was divided into six subpopulations, including Indica, Aus, Tropical japonica, Temperate japonica, Aromatic, and Admixtures [46]. These germplasms were planted in an experimental field in Nanjing for panicle blast inoculation. The seedlings of these germplasms were cultivated in 96-hole plugs in the greenhouse for seedling leaf blast inoculation. The rice blast isolate 2014-290B27 was provided by the Jiangsu Academy of Agricultural Sciences. The blast fungi were grown on corn rice straw agar plates at 28 °C for seven days and then transferred in black light (20 W) in an incubator at 28 °C for sporulation and culture for 7–10 days to promote spore production. They were then washed with sterilized distilled water to a spore suspension with a concentration of 1 × 10^5^ /mL [16,56].

### 4.2. Inoculation and Phenotypic Survey

In 2017 and 2018, natural population materials were planted in Nanjing, and inoculation was carried out at the booting stage. Each rice accession was inoculated with 9 rice panicles. The ear was sampled 15 days after inoculation, and the resistance survey was measured by calculating the percentage of the number of diseased grains in the whole panicle of rice (0–100%) [57]. For the leaf blast resistance identification of T-DNA insertion mutants and CRISPR plants of *Pb3*, the seedlings were planted in the greenhouse of Nanjing Agricultural University to the two-leaf and one-heart stage for spray inoculation [58]. The length and number of lesions were recorded after six days. Six plants for each line and three replicates were measured, and the average was calculated for the resistance phenotype. Suyunuo, a highly susceptible japonica rice variety, was planted around the tray as a control.

### 4.3. Population Structure Analysis and Genome-Wide Association Study

The phylogenetic tree of the 230 accessions was constructed using the MEGA7 program [59]. Multiple sequence alignment of 230 accessions was performed with the ClustalW program with the standard setting. The neighbor-joining (NJ) with maximum composite likelihood method was used to construct phylogenetic trees with a bootstrap value of 1000 replicates in MEGA 7. The beautification of the phylogenetic tree was conducted with iTOL online tools [60]. Based on the mixed linear model, GWAS was used to detect resistance-related loci [61]. The data of 400K SNPs (MAF ≥ 5%, missing rate < 25%) corresponding to the study accessions were filtered from the 700K high-density genetic markers of RDP-1 through the TASSEL 5.0 software. The MLM model was combined with the EMMAX online script (http://genetics.cs.ucla.edu/emmax/install.html, accessed on 20 December 2019) and R packages for analysis. Manhattan plots were drawn by using the R package qqman. *p*-value < 10^−4^ was used as the significance threshold to obtain the loci associated with leaf blast.

### 4.4. Candidate Genes and Haplotype Analysis

The LD heatmap of the associated loci in the GWAS was constructed using the Haploview4.2 [62]. Haplotype blocks were constructed using the confidence interval (r2 > 0.6) [21]. Candidate genes were predicted according to the MSU Rice Genome Annotation Project (http://rice.plantbiology.msu.edu, accessed on 10 June 2020). We analyzed the polymorphisms in the candidate region that were located on the promoter and caused amino acid changes. Haplotype analysis of these candidate genes was performed to screen out significant difference genes by using the Haploview4.2.

### 4.5. Genomic Nucleotide Diversity and Phylogenetic Analysis

To clarify the *Pb3* genetic differentiation of the natural population, the PLINK online script was used to extract SNPs on chromosome 11 in natural populations [63]. Then, nucleotide diversity (π) was calculated using VCFtools software [64]. Sliding 100 kb windows were used during the calculation with a 10 kb sliding step. The multiple sequences of R genes were aligned with MEGA7, then the phylogenetic trees were calculated and constructed using the neighbor-joining method within 1000 replicates. The conserved domain of the R gene was predicted by the Pfam online website (http://pfam.xfam.org/, accessed on 25 November 2021), and the gene structure was drawn by the IBS1.0.3 software [65].

### 4.6. RNA Extraction and qRT-PCR Analysis

Variety leaves infected with *M. oryzae* strains were collected at different time points after inoculation. The samples were immediately frozen with liquid nitrogen and stored in the refrigerator at −80 °C. The total RNA was extracted from leaves using TRIpure Reagent (Wuxi, China, Bio Teke Corporation, http://www.bioteke.com, accessed on 5 June 2021). First-strand cDNA synthesis was performed using HiScript II Q-RT Super Mix (Nanjing, China, Vazyme, http://www.vazyme.com, accessed on 6 June 2021). qRT-PCR was performed using the AceQ^®^ qPCR SYBR Green Master Mix. The primers in Appendix A were amplified for the genes’ expression analysis. The rice actin gene (*LOC_Os03g50885*) was used as the internal control. Each experiment was performed with three biological samples and each sample was assayed with three technical replications.

### 4.7. Generation of Transgenic Plants

The CRISPR-Cas9 gene mutation system was obtained from the Chen Qijun Laboratory, College of Biology, China Agricultural University, Beijing China [66]. The target primers were designed using the online website (http://skl.scau.edu.cn/primerdesign/, accessed on 12 September 2021). The CRISPR vectors pBUE411-2gR containing the *Pb3*-specific silencing target were transformed into japonica rice Bodao by an agrobacterium-mediated transformation method [67]. We obtained six CRISPR mutants of *Pb3* containing three editing types: type1:KO-1/ KO-2; type2: KO-3/KO-4/KO-5; type3: KO-6. The T-DNA insertion mutant of the Dongjin background of *Pb3* was purchased from Salk Institute Genomic Analysis Laboratory (http://signal.salk.edu/index.html, accessed on 11 July 2021). Sequencing results showed that an 8275 bp T-DNA was inserted into the exon of *Pb3* in the mutant. We obtained 5 homozygous mutant plants (pb3-1 to pb3-5) from 10 T_0_ generation seeds of the mutant. The primers in Appendix A were amplified for positive verification. 

## Figures and Tables

**Figure 1 ijms-23-14032-f001:**
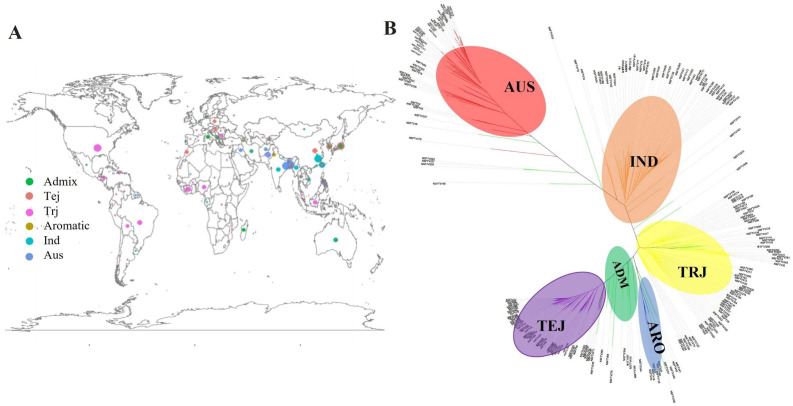
Population structure in O. sativa. (**A**) The large pie chart summarizes the distribution of subpopulations in the 230 O. sativa samples in our diversity panel, and the smaller pie charts on the world map correspond to the country-specific distribution of subpopulations sampled. (**B**) Phylogenetic tree of 230 accessions from the RDP1 in this study. ADM = admixture; ARO = aromatic; AUS = aus; IND = indica; TEJ = temperate japonica; TRJ = tropic japonica.

**Figure 2 ijms-23-14032-f002:**
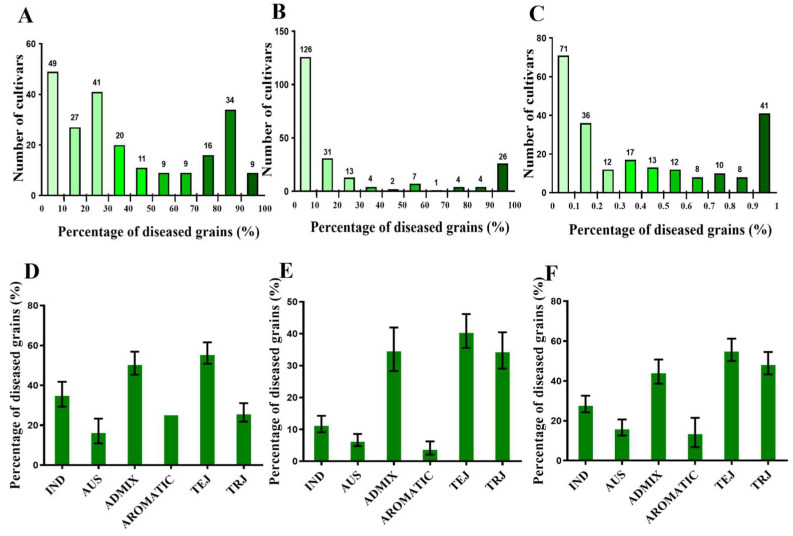
Phenotypic distribution of the 230 cultivars’ panicle blast resistance: (**A**–**C**) Diseased grains rate of the 230 cultivars by inoculating isolate 2014-290B27 in 2017, 2018, and 2019 in Nanjing, respectively. (**D**–**F**) Distribution of average diseased grains of the population in the six sub-populations.

**Figure 3 ijms-23-14032-f003:**
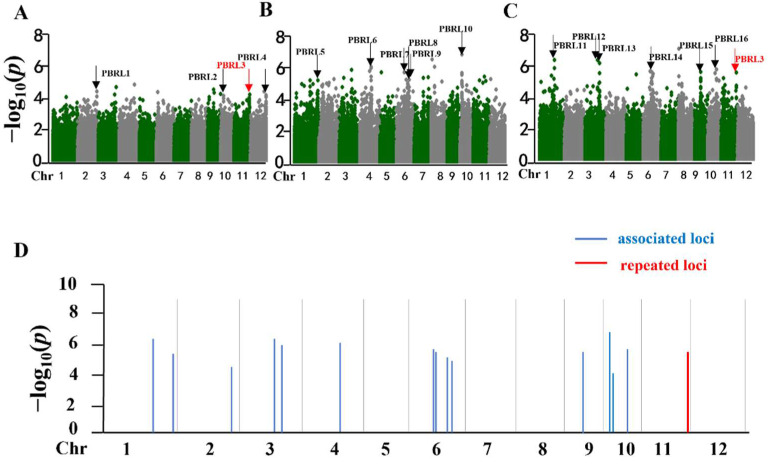
GWAS of the 230 cultivars’ inoculation and the distribution of rice-blast-associated loci. (**A**–**C**) Manhattan plots for panicle blast resistance in 2017, 2018, and 2019, respectively. The x-axis shows the single-nucleotide polymorphisms (SNPs) along each chromosome; the y-axis is the −log _10_ P for the association. The red arrows indicate the significant SNPs above the threshold, and the black arrows indicate the panicle blast resistance loci. (**D**) Summary of the identified loci of panicle blast. Each bar represents an associated locus. Blue and red lines indicate the associated loci and repeated loci, respectively.

**Figure 4 ijms-23-14032-f004:**
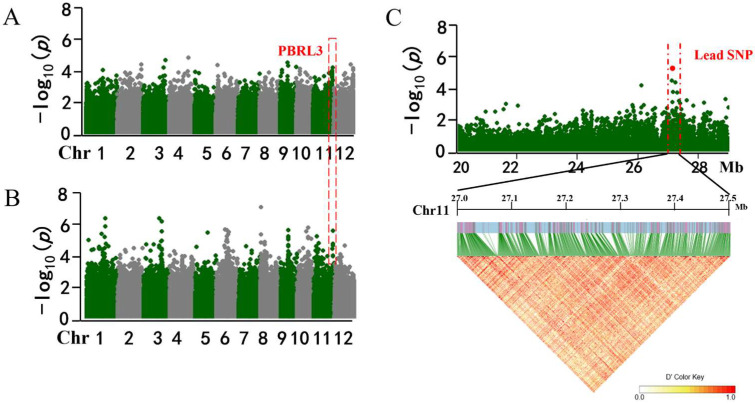
GWAS for leaf blast by inoculating the isolate 2014-290-B27 and LD block analysis for the peak on chromosome 11. (**A**,**B**) Manhattan plots from GWAS for rice blast resistance in June 2017 and 2019. The genomic coordinates are displayed along the x-axis and the logarithm of the odds (LOD) score for SNP is displayed on the y-axis. Different colored dots represent significant SNP obtained in different years. The red dashed box indicates the repeat site PBRL3. (**C**) Local Manhattan plot (**top**) and LD heatmap (**bottom**) surrounding the peak on chromosome 11. The red dot indicates the lead SNP at PBRL3.

**Figure 5 ijms-23-14032-f005:**
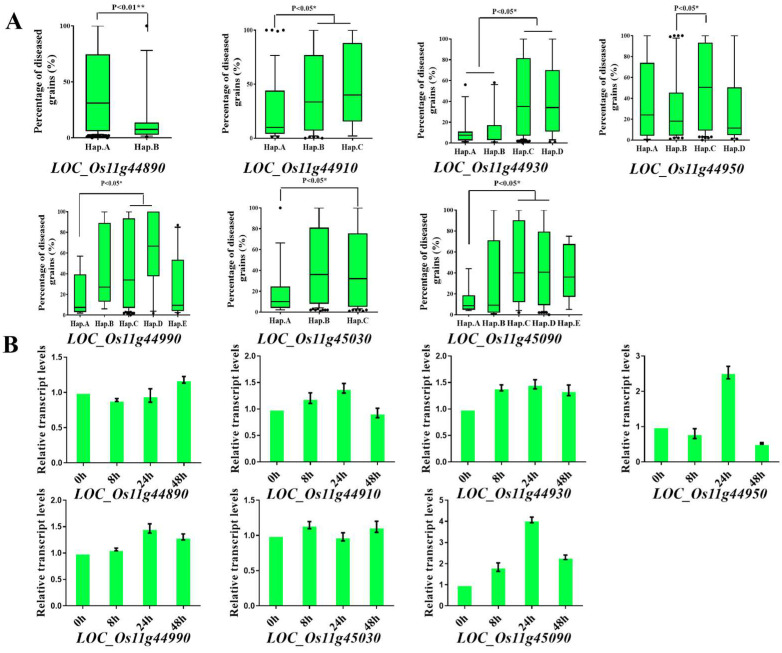
Candidate gene analysis of PBRL3 locus. (**A**) Haplotype analysis for candidate genes of PBRL3. (**B**) q-RT PCR analysis of nine potential candidate genes after inoculation. Data represent means ± s.d. (n = 3).

**Figure 6 ijms-23-14032-f006:**
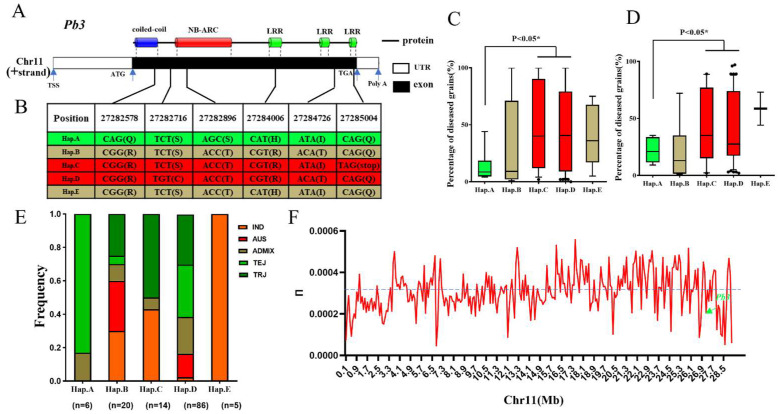
The haplotype of *Pb3* in the natural 230 population of RDP-1. (**A**) Gene structure of the candidate gene *Pb3* (**bottom**) and conserved domain in Pb3 (**top**). The blue, red, and green boxes represent different domains of PB3, respectively. The grey boxes represent the coding sequence of *Pb3*. (**B**) Haplotype analysis of the *Pb3* gene region based on the polymorphic sites shown in (**A**). (**C**,**D**) The distribution of phenotypes corresponding to different haplotypes in 2017 and 2019. (**E**) Frequency distribution of the five subpopulations in the five haplotypes of *Pb3*. (**F**) Nucleotide diversity of chromosome 11. The *Pb3* locus is indicated by a green arrow.

**Figure 7 ijms-23-14032-f007:**
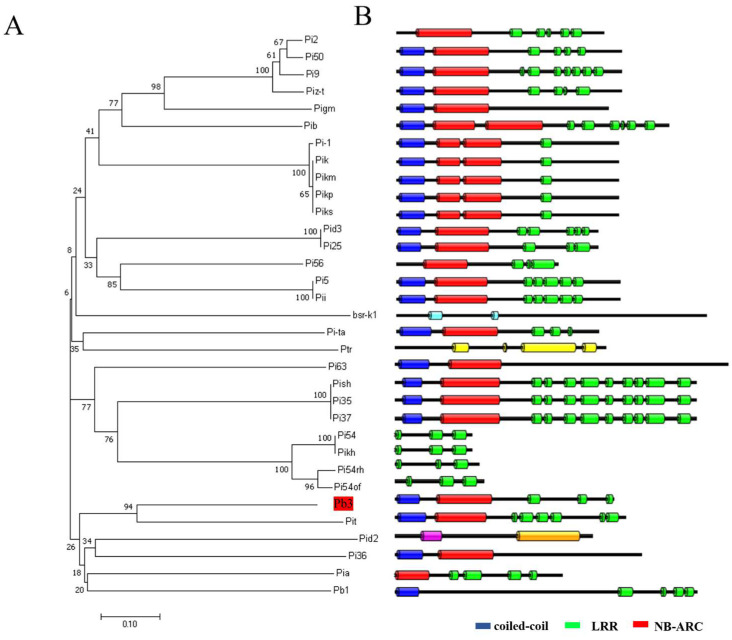
Evolutionary analysis of *Pb3*: (**A**) Phylogenetic tree was constructed based on the protein sequence of R genes by MAGE7.0. Pb3 is shaded in red. (**B**) The gene structure of R genes. Conserved domains were predicted via the Pfam online site. Amino acid sequences are represented by black lines and domains are represented by colorful boxes.

**Figure 8 ijms-23-14032-f008:**
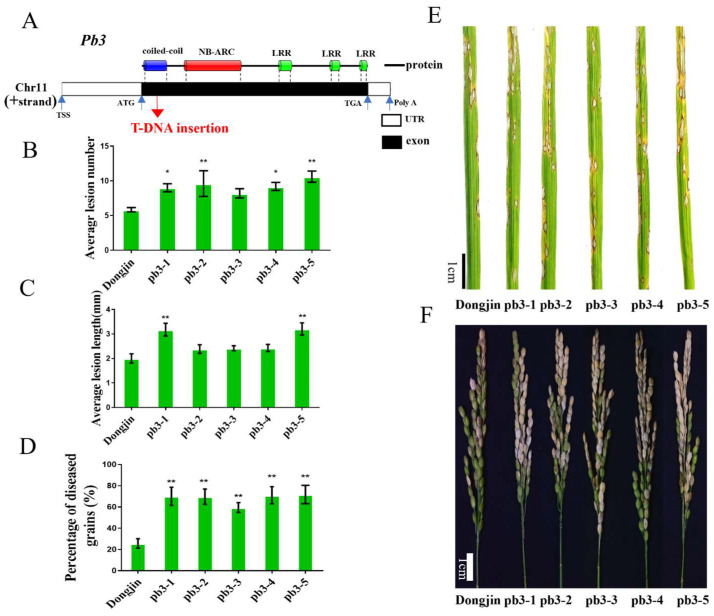
The phenotype of T-DNA insertion mutation lines of *Pb3*. (**A**) Schematic presentation of the gene structure of *Pb3* and T-DNA insertion sites. (**B**,**C**) The average lesion number and length in of wild-type Dongjin and five mutants inoculated with RO1-1 at the seedling stage. (**D**) Percentage of diseased grains of wild-type Dongjin and four mutant lines. Each material is inoculated with 9 panicles. (**E**) Spray inoculation of the four mutants and wild-type plants using strain RO1-1. (**F**) Injection inoculation at the booting stage of the four mutants and wild-type plants using strain RO1-1. * *p* < 0.05, ** *p* < 0.01.

**Figure 9 ijms-23-14032-f009:**
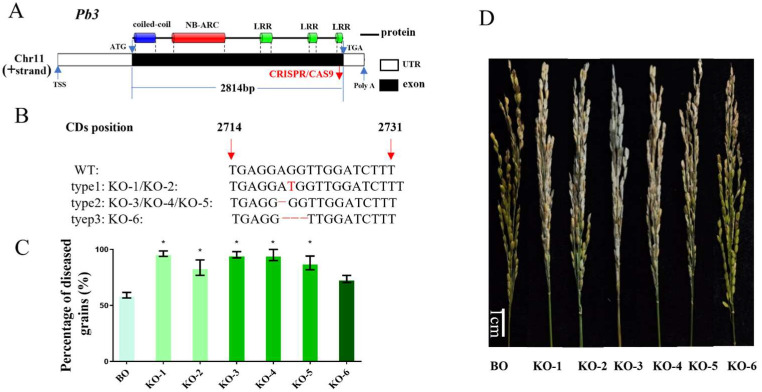
The Phenotype of CRISPR lines of *Pb3*. (**A**) Schematic presentation of the gene structure of *Pb3* and CRISPR/CAS9 mutation sites. (**B**) Mutations in the three knockout lines of *Pb3* in the Bodao background. The red arrow indicates the position of the CDs of *Pb3*. The bases marked red represent the inserted sequence in the genome. The red line indicates the missing sequence on the genome. (**C**) Percentage of diseased grains of the wild-type Bodao and three mutant lines. Different lines are distinguished by different colors. Each material is inoculated with 9 panicles. (**D**) Injection inoculation at the booting stage of the six mutants and wild-type plants using strain 2015-248A3. BO: Bodao. * *p* < 0.05.

**Table 1 ijms-23-14032-t001:** The PBRLs that were associated with panicle blast in three experiments by inoculating 2014-290-B27.

Locus	Chr	Position	Top SNP	*p*-Value	Years	Locus Reference
PBRL1	2	33611914–33651599	33611914	4.95 × 10^−5^	2017	*Pig(t)*
PBRL2	10	4834530–4844265	4834530	7.73 × 10^−5^	2017	
PBRL3	11	27114833–27391298	27118370	3.58 × 10^−6^	2017/2019	*Pik*
PBRL4	12	26529664–26995595	26954247	7.56 × 10^−5^	2017	
PBRL5	1	40778316–41195405	41195405	6.32 × 10^−6^	2018	LABR_19
PBRL6	4	20342333–20624109	20342333	8.79 × 10^−7^	2018	PBRL-14 *Pi-21*
PBRL7	6	13298661–13477930	13405216	1.88 × 10^−6^	2018	
PBRL8	6	20225421–20855898	20367124	8.57 × 10^−6^	2018	
PBRL9	6	21936925–22049270	21999647	1.59 × 10^−5^	2018	LABR_49
PBRL10	10	3525543–3631295	3582961	1.45 × 10^−7^	2018	
PBRL11	1	25009847–25590336	25472001	6.05 × 10^−7^	2019	LABR_11
PBRL12	3	21526136–22360496	21548437	6.22 × 10^−7^	2019	LABR_33
PBRL13	3	25404248–25800205	25404248	9.70 × 10^−7^	2019	LABR_34
PBRL14	6	13982991–14520587	14204045	3.33 × 10^−6^	2019	
PBRL15	9	10788559–10895765	10809843	3.13 × 10^−6^	2019	LABR_67
PBRL16	10	15103798–15256763	15251537	2.26 × 10^−6^	2019	

**Table 2 ijms-23-14032-t002:** The candidate genes of PBRL3 that were significantly different by haplotype analysis.

Locus	Chr	Gene ID	SNP Location	Gene Annotation	Haplotype Analysis
PBRL3	11	*LOC_Os11g44890*	CDs	expressed protein	*p* < 0.01
PBRL3	11	*LOC_Os11g44910*	Promoter CDs	DEAD-box ATP-dependent RNA helicase, putative, expressed	*p* < 0.05
PBRL3	11	*LOC_Os11g44930*	CDs	pentatricopeptide repeat domain containing protein, putative, expressed	*p* < 0.05
PBRL3	11	*LOC_Os11g44950*	Promoter CDs	glycosyl hydrolase family 3 protein, putative, expressed	*p* < 0.05
PBRL3	11	*LOC_Os11g44990*	CDs	NB-ARC domain containing protein, expressed	*p* < 0.05
PBRL3	11	*LOC_Os11g45030*	CDs	expressed protein	*p* < 0.05
PBRL3	11	*LOC_Os11g45090*	Promoter CDs	NB-ARC domain containing protein, expressed	*p* < 0.05

## Data Availability

Not applicable.

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
