# Peer review of "Genome-Wide Association Study Identifies a Rice Panicle Blast Resistance Gene Pb3 Encoding NLR Protein"

_ijms, 2022, doi:10.3390/ijms232214032_

Round 1

Reviewer 1 Report

Manuscript Number: IJMS-2004179

Title: Genome-Wide Association Study identifies a Rice Panicle Blast Resistance Gene Pb3 Encoding NLR Protein

Comments to Authors

This is a long-term project and the authors have spent a lot of effort on it. The study is beneficial for the plant sciences and really meaningful for agriculture and food security. However, several major concerns can be found in the manuscript. Based on these, I would like to suggest declining the current version of this manuscript for publishing in the journal and I am willing to re-review a new version of this work in the future.

1. Experimental design

- Lines 130-132, the authors should mention more detailed how they constructed the neighbor-joining analysis. Which factors and conditions were used here?

- Fig. 2A-C, it is better to show the actual number on each column.

- Fig. 5B, it is better to use “Relative transcript levels” for the y-axis.

- Fig.6A, 7B, 8A, and 9A, I am confused with the graphic picture displaying the gene structure here. It seems that the authors mixed between the DNA and protein structures? Black lines were used to show exons, there where were the introns missed?

- Regarding the T-DNA lines, the authors should clarify that these 5 lines (pb3-1 to pb3-5) are independent T-DNA insertion lines, or they all originated from 1 line? Based on the Supplementary Fig. S2, it is likely that all 5 lines are from the same line. Please describe in the Materials and Methods the information of T-DNA lines detailly. How long (kb) is T-DNA?

- A similar issue also found in the CRISPR-knock-out lines, the authors claimed that they got 6 lines. But the Supplementary Fig. S3 showed that lines 1/2 are the same while lines 3/4/5 are also the same. Please describe in the Materials and Methods the information on these lines detailly.

- A complementation assay is missed in the current study. I do understand that it is very challenging to generate transgenic rice lines. However, it is very important to include the complementation lines to solidly confirm that the Pb3 gene functions in the blast resistance.

- Primer and sgRNA sequences should be clearly provided in the Supplementary data.

2. Manuscript preparation

- Typos, English grammar, and other technical mistakes can be found in the manuscript. Some examples can be found as follows:

a. Line 65, “which has developed into” should be “which has been developed to be”.

            b. Line 71, “(SV),and”, a different font was used for the comma here and there is one extra space.

            c. Line 82, “28 445 SNP” is “28,445 SNP”?

- Abstract (line 19) and Introduction (lines 40, 44-46), please provide the full name of the Pb, Pi25, and Pigm/PigmR/PigmS genes in first use.

- Fig. 5 and Table 2, gene locus IDs should be displayed in italic form.

- Some sentences are hard to understand. For example:

            a. Lines 59-60, “In the field, a single R gene can usually provide 2-4 years of resistance in field cultivation.”.

Author Response

Response to Reviewer’s comments

Thanks for Editor’s comments. We have now carefully revised the manuscript and addressed all reviewers’ comments. We hope this revised MS (manuscript) could meet the standards for publication in IJMS.

Reviewer comments:

Reviewer #1

  1. Experimental design

- Lines 130-132, the authors should mention more detailed how they constructed the neighbor-joining analysis. Which factors and conditions were used here?

Answer: Thanks for reviewers’ comments. We added the sentence “Multiple sequence alignment of 230 accessions has been performed with the ClustalW program with standard setting. The neighbor-joining (NJ) with maximum composite likelihood method was used to construct phylogenetic trees with a bootstrap value of 1000 replicates in MEGA 7. The beautification of phylogenetic tree was conducted with iTOL online tools” in P20 line 406-410 in the revised MS.

- Fig. 2A-C, it is better to show the actual number on each column.

Answer: Thanks for reviewers’ comments. We added the cultivars numbers of each column in Figure 2A-C in P6 line154 in the revised MS.

- Fig. 5B, it is better to use “Relative transcript levels” for the y-axis.

Answer: Thanks for reviewers’ comments. We changed “Relative expression” into “Relative transcript levels” in Figure 5B in P10 in the revised MS.

- Fig.6A, 7B, 8A, and 9A, I am confused with the graphic picture displaying the gene structure here. It seems that the authors mixed between the DNA and protein structures? Black lines were used to show exons, there where were the introns missed?

Answer: Thanks for reviewers’ comments. In this study, the Pb3 contains one exon and none intron, and encodes an NBS-LRR protein with coiled-coil, NB-ARC and LRR domains. In order to distinguish the DNA and protein structures, we added the gene structure of Pb3 in Figure 6A in P12, Figure 8A in P15, and Figure 9A in P17 in the revised MS.

- Regarding the T-DNA lines, the authors should clarify that these 5 lines (pb3-1 to pb3-5) are independent T-DNA insertion lines, or they all originated from 1 line? Based on the Supplementary Fig. S2, it is likely that all 5 lines are from the same line. Please describe in the Materials and Methods the information of T-DNA lines detailly. How long (kb) is T-DNA?

Answer: Thanks for reviewers’ comments. We obtained only one Pb3 T-DNA insertion mutant line, and all five lines (pb3-1 to pb3-5) were produced from this line. We detected that the length of inserted T-DNA was 8275bp by sequencing. We added the details “Sequencing results showed that an 8275bp T-DNA was inserted into the exon of Pb3 in the mutant. We obtained 5 homozygous mutant plants (pb3-1 to pb3-5) from 10 T0 generation seeds of the mutant.” in P21 line 459-461 in the revised MS

- A similar issue also found in the CRISPR-knock-out lines, the authors claimed that they got 6 lines. But the Supplementary Fig. S3 showed that lines 1/2 are the same while lines 3/4/5 are also the same. Please describe in the Materials and Methods the information on these lines detailly.

Answer: Thanks for reviewers’ comments. We added sentence “We got six CRISPR mutants of Pb3 containing three editing types: type1:KO-1/ KO-2; type2: KO-3/KO-4/KO-5; type3: KO-6.” in P21 line 456-457 in the revised MS. We changed “6 homozygous CRISPR lines” into “3 homozygous CRISPR lines (type1, type2, type3)” in P14 line 271 in the revised MS. We also added the mutation sequence of three knockout lines in Figure 9B in P17 in the revised MS.

- A complementation assay is missed in the current study. I do understand that it is very challenging to generate transgenic rice lines. However, it is very important to include the complementation lines to solidly confirm that the Pb3 gene functions in the blast resistance.

Answer: Thanks for reviewers’ comments. We have constructed the complementary transgenic vector of Pb3 but it will take us long time to generate the transgenic plants. We hope to show the result in another paper in the future.

- Primer and sgRNA sequences should be clearly provided in the Supplementary data.

Answer: Thanks for reviewers’ comments. We added the primer and sgRNA sequences in the “Supplementary Table S1”.

  1. Manuscript preparation

- Typos, English grammar, and other technical mistakes can be found in the manuscript. Some examples can be found as follows:

  1. Line 65, “which has developed into” should be “which has been developed to be”.

Answer: Thanks for reviewers’ comments. We changed “which has developed into” into “which has been developed to be” in P2 line 68 in the revised MS.

  1. Line 71, “(SV),and”, a different font was used for the comma here and there is one extra space.

Answer: Thanks for reviewers’ comments. We corrected the wrong comma font and removed the extra space in P2 line 74 in the revised MS.

  1. Line 82, “28 445 SNP” is “28,445 SNP”?

Answer: Thanks for reviewers’ comments. We changed “28 445 SNP” into “28,445 SNP” in P2 line 85 in the revised MS.

- Abstract (line 19) and Introduction (lines 40, 44-46), please provide the full name of the Pb, Pi25, and Pigm/PigmR/PigmS genes in first use.

Answer: Thanks for reviewers’ comments. We changed “Pb1” into “Panicle blast resistance-1 (Pb1)” in P1 line 41-42 in the revised MS. However, the full name of Pi25 and Pigm are not frequently used in the paper, so we didn't change them in the revised MS.

- Fig. 5 and Table 2, gene locus IDs should be displayed in italic form.

Answer: Thanks for reviewers’ comments. We changed the gene locus IDs into italic form in Figure 5 and Table 2 in P11 in the revised MS.

- Some sentences are hard to understand. For example:

  1. Lines 59-60, “In the field, a single R gene can usually provide 2-4 years of resistance in field cultivation.”.

Answer: Thanks for reviewers’ comments. We changed sentence “In the field, a single R gene can usually provide 2-4 years of resistance in field cultivation.” into “In the field cultivation, Blast resistance mediated by single or a few major blast resistance genes (R genes) is easily overcome by M. oryzae ” in P2 line 60-62 in the revised MS.

Reviewer 2 Report

In the manuscript entitled “Genome-Wide Association Study identifies a Rice Panicle Blast Resistance Gene Pb3 Encoding NLR Protein”, the authors used GWAS to identify a rice panicle blast resistance gene, Pb3. Identification of the Pb3 gene was based on the panicle blast resistance phenotypes of 230 Rice Diversity Panel I (RDP-I) accessions with 700,000 single-nucleotide polymorphism (SNP) markers. A rice T-DNA insertion mutant in Pb3 was obtained. This insertion was located in the double coiled coil region of the gene and this mutant line showed susceptibility to panicle blast suggesting that the Pb3 gene is play a role in rice blast resistance. To confirm that Pb3 is indeed a blast resistance gene, the authors used CRISPR/Cas9 gene editing system to mutate the LRR domain. Homozygous CRISPR knock-out line showed increased susceptibility to panicle blast compared to wild-type plants. This demonstrated that the Pb3 LRR domain function in panicle blast resistance.

 The manuscript is well-written, and the analysis done is quite detailed. I could not detect any errors with the experiments or analysis present by the authors. The analysis and experiments support the conclusions made. However, some of the figures and legends are blurry or too small making them difficult to read. I would suggest that the authors consider increasing the clarity of the figures and legends.

Author Response

Reviewer comments:

Reviewer #2

In the manuscript entitled “Genome-Wide Association Study identifies a Rice Panicle Blast Resistance Gene Pb3 Encoding NLR Protein”, the authors used GWAS to identify a rice panicle blast resistance gene, Pb3. Identification of the Pb3 gene was based on the panicle blast resistance phenotypes of 230 Rice Diversity Panel I (RDP-I) accessions with 700,000 single-nucleotide polymorphism (SNP) markers. A rice T-DNA insertion mutant in Pb3 was obtained. This insertion was located in the double coiled coil region of the gene and this mutant line showed susceptibility to panicle blast suggesting that the Pb3 gene is play a role in rice blast resistance. To confirm that Pb3 is indeed a blast resistance gene, the authors used CRISPR/Cas9 gene editing system to mutate the LRR domain. Homozygous CRISPR knock-out line showed increased susceptibility to panicle blast compared to wild-type plants. This demonstrated that the Pb3 LRR domain function in panicle blast resistance.

 The manuscript is well-written, and the analysis done is quite detailed. I could not detect any errors with the experiments or analysis present by the authors. The analysis and experiments support the conclusions made. However, some of the figures and legends are blurry or too small making them difficult to read. I would suggest that the authors consider increasing the clarity of the figures and legends.

Answer: Thanks for reviewers’ comments. We adjusted the size of the legend in the figures and update the legends to clarity the figures.

Round 2

Reviewer 1 Report

Even though the authors did not supply the complementation assay in this work. However, the work contains T-DNA inserted line and CRISPR KO lines. Thus I believe that it is good enough at this moment. Other minor points were addressed all by the authors. I recommend to publish this nice work in IJMS.